# Teaching Online during the COVID-19 Pandemic: A Phenomenological Study of Physical Therapist Faculty in Brazil, Cyprus, and The United States

**Laura Plummer** [1], **Beliz Belgen Kaygısız** [2], **Cymara Pessoa Kuehner** [3], **Shweta Gore** [1], **Rebecca Mercuro** [1], **Naseem Chatiwala** [1] and **Keshrie Naidoo** [1,*]

1   Department of Physical Therapy, MGH Institute of Health Professions, Boston, MA 02129, USA; lplummer@mghihp.edu (L.P.); sgore@mghihp.edu (S.G.); rmercuro@bwh.harvard.edu (R.M.); nchallawala@mghip.edu (N.C.)
2   Department of Physiotherapy and Rehabilitation, European University of Lefke, Lefke, Northern Cyprus, TR-10 Mersin, Turkey; bkaygisiz@eul.edu.tr
3   Department of Physiotherapy, Centro Universitário Christus, Fortaleza, CE 60160-230, Brazil; cpkuehner@gmail.com
*   Correspondence: knaidoo@mghihp.edu

**Abstract:** The COVID-19 pandemic led to a global transition from in-person to online instruction leaving many higher education faculty with little time or training for this responsibility. Physical therapist education programs were especially impacted since a large part of the development of skills rely on face-to-face onsite practice. This phenomenological study explored the perceptions of physical therapist educators in three countries—Brazil, Cyprus, and the United States, who transitioned to an entirely virtual medium of teaching during the pandemic. Sixteen faculty participated in 1:1 semi-structured interviews. Trustworthiness of qualitative inquiry was ascertained using triangulation, thick descriptions, and peer reviews. Four major themes emerged from analysis of participants' interview data: adapting pedagogy in real-time, expected excellence, limitations of the medium, and informing future teaching practice. All participants described teaching during the pandemic as one of the most challenging experiences of their professional careers. Despite available resources, faculty noted challenges in making authentic connections with students, adapting to technological interruptions, assessment of student understanding of content, and managing work-life balance. Despite the challenges, faculty worked collaboratively with peers to innovate new approaches of creating social, cognitive, and teaching presence. Unique opportunities arose from the pandemic to enhance future teaching practice.

**Keywords:** physical therapist educators; online teaching and learning; pandemic

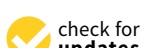

## 1. Introduction

On 30 January 2020, The World Health Organization Director-General declared the outbreak of novel coronavirus 2019 (COVID-19) "a public health emergency of international concern" [1]. In response to the outbreak, countries adopted safety measures, including social and physical distancing, travel restrictions, and stay-at-home orders [2,3]. Limitations of large gatherings resulted in educational institution closures worldwide, necessitating rapid transition from face-to-face academic instruction to online delivery. The pandemic has impacted an estimated 280 million learners across 22 countries, affecting over 80% of the global student population [4]. As the pandemic spread globally, school closures began in March in the United States [5] and Cyprus [6,7], and April in Brazil [8]. To avoid disruption of education, institutions of higher education rapidly transitioned to virtual learning formats leaving educators with little choice or time to prepare.

The transition to remote learning posed additional challenges to health professions programs, including physical therapy (PT). Since PT is a profession that leverages hands-on

skills [9], learner development of patient assessment and treatment skills requires a notable amount of training in a face-to-face setting [10–13]. An additional component of physical therapist education, like many health sciences curricula, includes an apprenticeship in the clinical environment. The concept of guided practice in the healthcare environment under the role of a mentor is one of the hallmarks of PT education. Classroom knowledge does not begin to make sense to the PT student until they apply their knowledge and skills in the clinical environment and the student engages with the community of practice [14]. However, because of the pandemic, PT students were forced to leave their classrooms and clinical sites due to limitations on non-essential personnel in healthcare facilities.

Before the pandemic, the delivery of PT education in the US was evolving to include online, blended, and hybrid learning experiences to minimize barriers to education. Deliberate curricular design, which leverages both online (synchronous and asynchronous) and face-to-face learning strategies to maximize learning in both environments, continues to gain traction in the health professions education, including PT [11–13,15]. However, despite the growing research, implementation of fully hybrid or blended PT programs that employ online instruction, followed by intensive onsite coaching in laboratory classes to solidify and assess motor skills, are still scarce within the US, and non-existent in Brazil and Cyprus. Unlike the deliberate development of a blended or hybrid curriculum in PT programs pre-pandemic, teaching during the pandemic required a quick pivot, in some cases within a matter of days, with little time for extensive faculty training in technology or online pedagogy. While the impact on students during the pandemic cannot be overstated [3,16–23], physical therapist faculty across the globe were challenged to convert traditional in-person curricula, heavy in psychomotor skill development, to entirely virtual models to keep students safe, socially distanced, and on track for graduation.

The pandemic provides a unique opportunity to explore the viability of virtual PT education and the associated burden incurred by physical therapist educators. To date, this experience has not been investigated. Therefore, the purpose of this phenomenological study was to explore the perceptions of physical therapist educators in Brazil, Cyprus, and the United States, new to online teaching, who transitioned their in-person curricula to entirely virtual models during the COVID-19 pandemic. We selected three countries, separated by thousands of miles, to investigate the barriers and facilitators to remote PT education in these countries but also to explore the shared experience of this phenomenon and how lessons learned from remote instruction might inform PT educators' future teaching practice.

*Theoretical Frameworks*

As described, the faculty in this study were new to online teaching. The COVID-19 pandemic forced institutions of higher education to consider, not whether quality education could be delivered virtually, but how to implement online teaching and learning in a short period of time. Even prior to the pandemic, natural disasters necessitated e-learning in many parts of the world. Dhawan [24] highlights that online learning was the future but that the pandemic accelerated the process. However, there is a difference between online learning that is implemented with time to orient educators to the pedagogy of online learning and the crisis e-learning, which was implemented by the faculty in this study. In describing the rapid shift to online learning during the pandemic, or crisis e-learning, Dhawan coins the term "*Panicgogy*" [24]. Despite being unfamiliar with online learning pedagogy, faculty may have put existing frameworks into practice. As such, two theoretical models situate this study. The first is the online community of inquiry (CoI), which uses the themes of social, cognitive, and teaching presence to maximize online learning [25]. Social presence establishes relationships between faculty and students and between students through effective and open communication. Cognitive presence explores, constructs, and confirms understanding through collaborative group work within a CoI. Teaching presence involves the design, facilitation, and direct instruction within the CoI to work towards

learning objectives [26]. We sought to investigate which elements of a CoI faculty leveraged when they transitioned their curricula to a virtual platform.

The second theory, the cognitive apprenticeship model (CAM), addresses the challenge of providing effective and meaningful feedback through open communication and cohesion in an online classroom [27]. Cognitive apprenticeship involves the learner completing tasks in the authentic environment with the focus shifted on expert reasoning, which precedes and occurs during task completion [28–30]. The purpose of cognitive apprenticeship is to bridge the gap between knowledge taught in the classroom and the application of knowledge in the real world [28,30]. Cognitive apprenticeship is vital to facilitate PT students transferring classroom concepts to a dynamic and complex clinical environment. With the transition to online learning, physical therapist students were missing the opportunity for apprenticeship in the clinical environment. We sought to explore how PT faculty fostered cognitive apprenticeship in the virtual environment.

## 2. Materials and Methods

### 2.1. Study Design

Phenomenology is a careful examination of human experiences focused on how people make sense of their engagement in the world and life experiences. This qualitative inquiry method also focuses on making meaning of participants' experiences with a shared phenomenon [31,32], which in this case is a transition to online teaching. We used a comparative phenomenological design to investigate how physical therapist faculty from three countries perceived the transition to online learning during the pandemic, discover their perceived barriers and facilitators to delivering physical therapist education online, and explore how the transition to online learning may inform teaching practices moving forward once no longer mandatory.

### 2.2. Participants and Context

Three higher education institutions participated in this study: Institution A in the United States, Institution B in Brazil, and Institution C in Cyprus. Physical therapist education in all three institutions occurred primarily in-person before the pandemic. Institution A is a graduate school in the Northeast region of the United States, dedicated to the health sciences with entry-level and post-professional programs in PT, occupational therapy, physician assistant studies, speech-language pathology, nursing, and genetic counseling. At the time of this study, the PT program included 211 graduate students, supported by 22 core faculty members. A typical PT class in Institution A comprises 70 students. PT students in Institution A complete two years of didactic coursework followed by a year-long clinical internship.

The physiotherapist degree requirements in Institution B involve completing a five-year undergraduate program, including one year of clinical internship. Institution B, located in Northeastern Brazil, includes both undergraduate and graduate programs. The 21 undergraduate programs include those dedicated to the health sciences (medicine, physiotherapy, nursing, dentistry, biomedicine, psychology, and nutrition). At the time of the study, Institution B included 360 physiotherapy students supported by 34 faculty members. Class sizes average between 30–40 students.

In Cyprus, a physiotherapy degree requires a four-year bachelor's degree that includes health and field-specific coursework. Institution C, located in North Cyprus, offers both undergraduate and graduate programs. Undergraduate programs are taught by 11 faculty including those in the health sciences department. The Institute of Graduate Studies and Research at Institution C offers both Master's in Science and Doctor of Philosophy degrees. At the time of this study, Institution C included 328 students in the physiotherapy department supported by nine faculty. The average class comprises 80 students. Students complete theory-based courses during the first three years of study, followed by a year-long clinical internship.

### 2.3. Sampling

Researchers used purposeful sampling to recruit faculty from three major continents—North America, South America, and Europe, that were geographically distanced to make our results globally representative. Additionally, we identified institutions of higher education from the US, Brazil, and Cyprus where the physical therapy curricula were taught primarily in-person prior to the pandemic. The transition to virtual learning began between March and April in these countries and all three academic institutions had comparable infrastructure and resources. Faculty members who were full-time or part-time employees in the PT department, who had delivered at least one course online during the pandemic with a minimum of 24 h of synchronous online teaching over a twelve-week semester were asked to participate in virtual 1:1 interviews with researchers. The need to speak English was not an inclusion criterion as researchers in Institutions B and C conducted interviews with faculty in their first language of Portuguese or Turkish.

### 2.4. Ethical Approval

Institutional review board approval was obtained from each of the three participating institutions. All participants reviewed an information sheet and the interview protocol before participating in a 1:1 virtual interview. Participants were informed that researchers would deidentify all study data and report only aggregate data. The researchers specified criteria for stopping data collection to safeguard participant time, including when data saturation had occurred (no new codes or themes emerged from the data during concurrent thematic analysis).

### 2.5. Instrumentation

Researchers from all three institutions participated in designing the interview protocol. Interview questions were based on the protocol used by Oreshkina in a phenomenological study exploring teacher experience in three countries (South Africa, Russia, and the United States) [33]. The final semi-structured interview protocol included three consent questions, six open-ended questions, and seven demographic questions (Appendix A). The brevity of the interview protocol helped to keep the focus on the participants' experience of the phenomenon under study (transitioning to online teaching during the pandemic) [34]. As the interviews were conducted by different researchers with participants in three countries, researchers also specified common follow-up questions and prompts. Once finalized, the interview protocol was translated to Portuguese and Turkish by researchers for use at Institutions B and C.

### 2.6. Data Collection and Analysis

To minimize coercion between the researchers and fellow faculty members, program staff at each institution distributed recruitment materials via email to faculty. Due to a larger local sample (n = 8) who consented into the study at Institution A, two faculty from that institution conducted 1:1 interviews with those participants. One faculty from Institution B and one faculty from Institution C interviewed faculty from their institutions. Virtual interviews, each lasting a maximum of 45 min, were conducted during October and November 2020 using Zoom (Institution A), Google Meet (Institution B), or Microsoft Teams (Institution C). Interviews were audio-recorded and transcribed. The four researchers shared their field notes and transcripts to allow for concurrent data analysis. Data analysis concurrent with data collection allowed for correcting blind spots throughout the study period and to ensure that sample and variable saturation was approached by the time the last interview was conducted [35].

Each researcher who conducted the interview reviewed the interview transcript for accuracy. Transcripts from Institutions B and C were translated into English before thematic analysis commenced. Transcripts and field notes were subjected to the six-step process for thematic analysis by two researchers using NVivo Software, QSR International Pty Ltd. The six steps include: (1) familiarizing themselves with the data, (2) generating initial

codes, (3) searching for themes, (4) reviewing themes, (5) defining and naming themes, and (6) producing the report [36]. During step 1, each researcher read the data multiple times for initial ideas. During Step 2, the researcher completed first cycle coding [35]. The researchers relied on a descriptive coding process and summarized data chunks into words or short phrases (see Table 1 for examples of codes). The researchers used an inductive coding approach due to the lack of research into the area of virtual physical therapy education during the pandemic. During step 3, the researchers met to review and compare the codes generated and agree on the final codes and operational definitions. The researchers reached 85% agreement on codes which is within the acceptable range of agreement [35]. The researchers jointly completed second cycle coding at this stage. During second cycle coding, the codes were grouped together into smaller categories (themes) and assigned names (see Table 1). All four researchers who completed the interviews completed step six. Step six included identifying compelling quotes from participants across contexts.

**Table 1.** Themes and supporting codes.

| Theme | Codes | Description of Code | Code Supported by Data from Institution A, B, or C |
|---|---|---|---|
| Adapting Pedagogy in Real-Time | Steep learning curve | Orienting to new technology and pedagogy in an abbreviated period | Institution A, B, and C |
| | Multiple moving parts | Teaching online involved instruction, monitoring technology, and the student experience resulting in divided attention (for both faculty and students) | Institution A and C |
| | Overwhelmed | Adapting curricula and pedagogy while stressed about the pandemic | Institution A, B and C |
| | Innovation | Both faculty and students had to be creative to teach and learn psychomotor and patient care skills online | Institution A, B, and C |
| | Resources | Faculty relied on instructional designers/pedagogical support centers, information technology support, and external support | Institution A, B, and C |
| | Collaboration | Teamwork and support from fellow faculty seen as the greatest resource | Institution A, B, and C |
| Expected Excellence | High standards | Faculty used to being experts were now novices in teaching online | Institution A |
| | Expectations | Faculty were managing expectations from leadership and students | Institution A and B |
| | Psychosocial stressors | Workload, lack of rest, concern for student wellbeing | Institution A and B |
| | Caring | Invested in the educational experience/product. Motivated to keep students on track to graduate | Institution A, B, and C |
| Limitations of the Medium | Teaching psychomotor skills online | Hands-on nature of PT education in the virtual space resulted in missing subtleties and a lack of control | Institution A, B, and C |
| | Missing connection | Faculty felt disconnected from fellow faculty, from students, and felt that students missed opportunities for peer learning | Institution A, B, and C |
| | Lack of teaching feedback | Missing the in-the-moment feedback from students to inform the teaching process | Institution A |
| | Assessing understanding | Faculty questioned whether students were understanding content delivered, "Did they get it?" | Institution A, B, and C |

**Table 1.** *Cont.*

| Theme | Codes | Description of Code | Code Supported by Data from Institution A, B, or C |
|---|---|---|---|
| | Low student engagement | Students not engaging with material, faculty concern about knowledge retention | Institution A, B, and C |
| Informing Future Teaching Practice | Flexibility of online teaching and learning | Students benefited from learning at their own pace and taking time to process | Institution A and C |
| | Unique opportunities | Introducing students to telehealth, use of the 1:1 coaching model for feedback | Institution A and B |
| | Establishing different types of connection | Online learning broke down barriers and the power differential between students and faculty | Institution A, B, and C |
| | Gratitude | Faculty were grateful to be have been able to retain their jobs | Institution B |
| | Grit | Faculty gained insight into their resilience when faced with adversity | Institution A and B |

*2.7. Trustworthiness*

Several methods, including member checking, triangulation, thick descriptions, peer reviews, and external audits, are available to work toward validity in qualitative inquiry [37]. In this study, researchers leveraged two forms of triangulation: data triangulation and researcher triangulation, as well as peer review and thick descriptions to increase trustworthiness. Data triangulation included analyzing interview transcripts and field notes from four researchers. Researcher triangulation leveraged data analysis performed by two of the researchers with experience in qualitative research. Secondly, researchers from all three countries reviewed the codes and themes for accuracy. Researchers also ensured that the results represented the data set by selecting quotes from participants across the three institutions. Finally, as the researchers adopted a constructivist paradigm during thematic data analysis, thick, rich descriptions were used to work towards credibility in this analysis [38,39].

**3. Results**

Sixteen faculty members participated in 1:1 interviews with researchers: eight in the US, five in Brazil, and three in Cyprus. All faculty members were licensed physical therapists in their country who had, on average, eight years of experience as educators (range = 1.5–18 years). Only one participant reported experience teaching online before the COVID-19 pandemic, and ten faculty (62.5%) reported having experience as students in the online learning environment. Researchers identified the following themes following thematic analysis of interview data: adapting pedagogy in real-time, expected excellence, limitations of the medium, and informing future teaching practice.

There were a few instances where a phenomenon was only experienced by faculty at one of the institutions. Participants in Institution A described the high standards set in their context and the discomfort associated with feeling like novices in the online teaching arena (see Table 1). The phenomenon was not noted at the other two institutions. Similarly, every participant from Institution B reported being grateful that they were able to retain their jobs while many in the country faced unemployment, an experience unique to the context of Institution B. However, ultimately, there were more similarities than differences during the phenomenon under study (as evidenced by Table 1). Each theme is discussed further next.

*3.1. Theme 1: Adapting Pedagogy in Real-Time*

With little time to adapt, participants described a steep learning curve associated with converting classes, designed to be delivered in-person, to a fully online format. In some

instances, faculty had as little as ten days' notice (in Institutions A and C) before country-wide social distancing measures were put in place and teaching transitioned to fully remote. One participant in Institution A described the transition as a whirlwind, another as: "*trying to fly the plane while writing the manual at the same time.*" Needing to master multiple pieces of new technology was described by some as anxiety-provoking, and faculty relied on a variety of technology with preference for medium varying by country.

While participants were aware of the challenges facing all educators teaching virtually, they highlighted the uniqueness of PT curricula, which involve teaching a significant amount of psychomotor skills. Converting content that was typically delivered in-person required innovation and creativity from faculty and students. Additionally, students in all three countries were residing in different time zones, sometimes socially distancing alone, in varying home situations. To overcome the challenges of teaching psychomotor skills online, participants in Institution A described using video resources to demonstrate a skill followed by having students practice the skills on family members or roommates. Students would then film themselves to receive 1:1 feedback from faculty. Students who were social distancing alone also demonstrated creativity with finding solutions to practice and demonstrate their skills:

> *Some of the more memorable ones were watching students using stuffed animals to be their patients, or in one instance, using their pet dog to do elbow and wrist range of motion, and it was incredibly creative and sweet and probably much better than not having a model at all. (Participant, Institution A)*

Participants used a host of technological resources and games to foster student engagement. One participant in Institution A described mocking up their home to allow students the opportunity to practice an in-home evaluation with a patient. Performing patient telehealth visits with students required authorization from the physiotherapy council in some countries. However, once approved, participants described the student and patient interaction as incredibly rewarding. Here a participant describes a patient's reaction to the students' efforts to provide treatment virtually:

> *She was talking about her functional limitations during her daily activities when she stopped to thank our group for the effort we were all making to see and talk to the patients and try to provide them with some sort of treatment. My students cried, and that made me realize that as hard as this pandemic can be, we can do something to help others. (Participant, Institution B)*

Transitioning to virtual instruction was also resource-intensive. Whether participants referred to instructional designers in the U.S. or pedagogical support centers in Brazil, the support of those with expertise in designing teaching and learning experiences was vital. Participants attended workshops and sought technological support and external resources. In the U.S., participants relied on faculty familiar with teaching online in hybrid PT programs. However, in all three countries, the most valuable resource was ultimately the collaboration and teamwork among the faculty at each institution. One participant from Institution B described, "*Teaching through this pandemic has been the hardest thing that I had to go through in my life, professionally, but it can be done efficiently if we work together as a group, learning with each other.*" Another participant in Institution A concurred, "*I found faculty's experience actually was probably the best part of the collaboration, the most useful collaborative resource. People who had the experience and were able to give us tips. What worked, what didn't work.*"

*3.2. Theme 2: Expected Excellence*

Participants in all three contexts stressed the exorbitant amount of time required to adapt their pedagogy in real-time and orient the rest of the teaching team. While participants felt supported by their institutions, some noted leaderships' lack of understanding about the amount of effort it would take to convert PT curricula to an online format in a short amount of time. Preparation time increased significantly. In the U.S, participants

described that laboratory instructors, usually charged with guiding student mastery of psychomotor skills in-person, were now serving as facilitators in the online environment. Both laboratory instructors and students were unfamiliar with the technological tools being employed. Students needed additional guidance on how to access materials and submit assignments on new platforms. A participant from Institution C described, "*It was a difficult process for us to get used to the system, for students to get used to the system, and to change the ways of education methods to remote instruction.*"

Faculty participants in all three countries described being conscious of multiple moving parts when delivering education in the virtual environment. As they shared their computer screen, taught their content, fielded questions from students, and attempted to monitor student engagement and use breakout room functions proficiently, participants were also assisting students with technological challenges. Teaching, already so nuanced, took on a new layer of complexity. A participant from Institution A recalled:

> *It was certainly stressful because, at any moment, Zoom could go off, a student could be disconnected, they would be stressed because they felt they missed important material. So you're trying to get back into Zoom, fielding texts because they're out of Zoom, but you're still trying to teach the content at the same time. There were just so many moving parts happening at the same that it was really stressful and fatiguing.*

Participants described being invested in the student educational experience and wanting to keep students on track to graduate. Faculty also cared about the educational product that they were creating. Some participants questioned whether they had tried to accomplish too much in a short period but were used to setting high standards for themselves. At times, participants working 14- and 16-h days questioned whether they were working from home or living at work. Some participants described the experience as demoralizing. They felt that they were investing a significant amount of time into creating an educational product that could be perceived as inferior. Participants, accustomed to serving as high-functioning classroom authorities, now felt pressure to advance their novice skills as online educators to expert levels in a short period. When participants observed role models who appeared to be functioning at a high level, they wondered about the costs. They described the experience as overwhelming, challenging, and exhausting:

> *Faced with this gigantic task and I have no clue how I'm going to do it and knowing that it's not an area that I'm comfortable in, the technology part of it . . . exhaustion was probably the biggest emotion. I was too tired to cry. (Participant, Institution A)*

While there was pressure to maintain high institutional and personal standards, the real driver was that participants cared about the student experience and expectations and were thinking about the ultimate stakeholder in this situation: the students' future patients:

> *You know, it's such a hard thing because I feel like no matter what mode we are in during this sort of pandemic, I think the psychological costs are high because you realize that things are different. We care a great deal about our students, them as humans but also knowing that they get what they need and deserve for the time they're putting into this and ultimately caring for others. (Participant, Institution A)*

*3.3. Theme 3: Limitations of the Medium*

The majority of participants, regardless of country, remarked on the challenge of low student engagement in the online environment. However, participants were acutely aware that students were stressed about the pandemic's uncertainty and unknown timelines. Students were understandably anxious to return to campus for in-person learning. Participants were also aware of the added pressures that students were facing. Participants in Brazil worried that students would not be able to afford to complete their programs with parents facing unemployment due to the pandemic. Even students who were able to remain enrolled were frequently dealing with a lack of access to technology or poor internet connections. Every participant reported being concerned about the effects of low student engagement and worried about students understanding and retaining the course

materials. One participant from Institution C remarked, *"While teaching online, I constantly questioned whether they understood the lesson. I think my questioning was about ten times more than the class situation."*

Participants described using no- and low-stakes assessment techniques much more frequently to overcome the barrier of low student engagement and held regular virtual open office hours. However, in this, they perceived an added limitation of the medium. Whereas students tended to gather around professors before and after class to talk in-person, seek clarification, and ask questions of an organic nature, participants noted that students only attended virtual office hours if they had a specific problem. They described that the organic discussions, common in the classroom or during informal meetings in offices and hallways, failed to materialize. Ultimately, this element of missing connection was perceived as the most significant limitation to teaching and learning in the virtual classroom.

All participants described that they missed the in-person connection with students. They noted a social awkwardness to the virtual platform where everyone except the speaker was muted. Additionally, privacy and student video-sharing issues added to the complexity of establishing a connection in the online environment. Participants felt that students should not feel compelled to share their videos and were conscious that some students did not feel comfortable sharing their home environment. One participant remarked:

> *Sure, the students tend to be more shy in virtual classes, and I think it's because they are in their home environment. So, they don't talk much and even turn off their cameras so that they don't have to show their background, their home, and how they live. (Participant, Institution B)*

However, participants emphasized that teaching requires real-time feedback or re-actions from students to assess understanding and inform the teaching process. They described it as challenging to teach to multiple muted, black screens. Participants described asking questions, telling jokes, and hearing "crickets":

> *I think the feel of the classroom, the general vibe of how your students are doing, and what you're able to ascertain just by being in the same room with them . . . gives you the opportunity to adjust how you're teaching or maybe what you're going to go to next, how you're going to set up a lesson, and that seems very hard to do in the virtual environment. There's much less of that feeling you get from your students. (Participant, Institution A)*

Participants also noted that students had limited opportunities to connect with class-mates thus missing opportunities for peer learning. To attempt to minimize this limitation, participants leveraged small group activities and virtual breakout rooms with facilitators when possible. However, virtual breakout rooms meant that faculty could not gauge the temperature of the room during small group activities and remarked that virtual teaching had an unfamiliar rhythm and pacing to which they had to acclimate. Finally, participants also missed their connection with their fellow faculty. While participants relied on innova-tion and creativity to teach psychomotor and patient care skills virtually, they remarked that PT curricula were ultimately not meant to be taught entirely online. Participants remarked that they were challenged to fit their content to the medium instead of the other way around. One participant from Institution A mentioned: *"For some reason, being on zoom encouraged us to want to lecture, and I think that was not what the students needed. It tired them out tremendously."* Participants felt that they were limited in their ability to introduce different patient care scenarios so that students would have to reason in the moment and make decisions about how to proceed with patient care and put psychomotor skills to use. Another participant from Institution A remarked, *"It was very limited what we could ask them to do, so we really couldn't introduce those higher-level skills."*

### 3.4. Theme 4: Informing Future Teaching Practice

Participants remarked on the flexibility that online teaching and learning afforded both students and educators when asked what lessons they might carry forward once they returned to in-person teaching. They described the benefit of students having time to learn

and process foundational concepts before coming to class to apply and practice content: "*They actually had time to process some of the cognitive information and sort of let it sit for a little bit and then perhaps their practice was more meaningful and at a higher level*" (Participant Institution A). Another participant concluded:

> *I never thought that this method would be suitable for physiotherapy education. I was thinking that online education may be appropriate for some courses in other departments, but not for physiotherapy. But for now, with the support of resources that our university provided us, I believe that this method can be used for some lectures in our profession as well. Now we are doing "hybrid" education . . . We have very positive feedback from students. (Participant, Institution C)*

There were unique opportunities for learning with the use of video technology and online coaching models. Participants gained an appreciation for the importance of allowing students to learn about and practice telehealth skills. While there was still the emphasis on missed connection when using virtual platforms, participants noted that they were able to make different types of authentic connections with students. Teaching and learning in their respective home environments, participants described having opportunities to interact with students' families, and students got to see their faculty's family members. Despite the physical separation, participants described feeling closer to students. In some ways, the virtual learning platform helped to decrease the power differential between students and their professors, as articulated here:

> *In the middle of the class, one of my kids, the youngest one, came inside my office asking for food, and the oldest came right after yelling at him because he was disturbing me. I hugged them both and calmed them down, excused [myself] for one second while I gave him some food and took them both to the living room. When I came back to class, the students were all smiling and said that they were touched by my kindness and calm [demeanor] with my kids. This was especially nice because my relationship with these students was very cold from the beginning, and all of a sudden, it changed after this. We became much closer, and they started to show up more for the classes. (Participant, Institution B)*

Another participant mentioned:

> *Being engaged all day made students share all their daily life activities and special occasions on the [online] platform. I had a student who wanted to celebrate her birthday after the lecture ended, which was a memorable experience for me . . . She said that she is away from friends and wanted everybody to write some [messages] for her. (Participant, Institution C)*

The final aspect of lessons learned from virtual instruction during the pandemic included gratitude for job security and an insight into their resilience and grit. Participants in Brazil described being relieved that they were able to retain their jobs while many in the country were facing unemployment. However, despite feelings of gratitude, all participants described teaching during the pandemic as the greatest challenge of their professional career and some had considered retiring from teaching. However, realizing the need for a well-prepared future generation of physical therapists and their roles as educators, they were motivated to persevere:

> *I learned that we have to be resilient. This process has been very difficult on everybody's lives, and we have to deal with these difficulties in order to keep going. I thought of giving up and retiring from teaching, but I saw everyone in the same situation and thought to myself that I could not give up. My reason for teaching is the pleasure that I have of seeing so many new good professional PTs, so I decided that no matter what happened, I had to continue. (Participant, Institution B)*

## 4. Discussion

In March 2020, the COVID-19 pandemic necessitated global safety measures of social and physical distancing and stay-at-home orders [2]. There was a sudden pause in the

work and life routine as we knew it. This was by far the most significant shift of the workforce to a remote work environment, affecting businesses and industries across the world. Faculty in physical therapist education programs found themselves needing to rapidly adopt new pedagogical models. Despite the faculty workload and new technology skills required for this transition to online learning, research in higher education during the pandemic has focused primarily on the student experience [17,19,20,22,23] and the effectiveness of virtual formats for student learning [16,18,21]. Faculty experience has been less thoroughly investigated [40–42]. This study describes the experiences of an international group of physical therapist faculty whose work-life shifted significantly during this global crisis. To our knowledge, this is the first study exploring the perceptions of physical therapist educators about the online transition. Notably, we observed more similarities than differences in the experiences of barriers and facilitators among this international group of faculty participants.

Preserving high-impact educational practices such as experiential learning [43] was particularly challenging with a transition to fully remote learning. Many educational activities, including service-learning projects, clinical education experiences, and internships, needed to be postponed. In all three countries, physical therapist faculty described several challenges with this sudden, unanticipated transition including steep learning curves and significantly increased workloads associated with meeting student needs. Changes from onsite to remote teaching could not occur without substantial changes to delivery methods, content, and learning assessments, thereby mandating more time and effort be allocated to preparing for teaching [12,44]. These substantial changes at such an accelerated pace came at a psychological cost to faculty who set high standards for themselves and were heavily invested in producing a quality educational product for their students. Faculty described virtual teaching as "living at work," often putting in 14–16-h days to design new learning activities and assessments and provide additional time dedicated to student support.

The community of inquiry theory identifies the importance of social, cognitive, and teaching presence for successful online learning [25,26]. Faculty described challenges to all three of these critical components of online learning. Social presence is the ability of participants to project themselves socially and emotionally as 'real' people through the medium being used, engaging in open communication and developing interpersonal relationships [25]. Faculty described struggles with creating social presence as students joined class sessions from home, sometimes uncomfortable sharing their screens and thus their home environments. While faculty appreciated and respected students' need for privacy, they reported challenges connecting with students who chose to keep their cameras off. Mood and tone, traditional indicators of presence and engagement, were lacking within the virtual environment where only the speaker was unmuted. These findings were consistent with a previous study of faculty in the Philippines teaching online during the pandemic who reported ambivalence with online education due to feelings of depersonalized education [40].

With students joining class from their homes which were often in different time zones, supporting relationship-building was imperative. Although faculty leveraged virtual breakout rooms for small group discussions to promote personalized education, this strategy required additional laboratory instructor training in pedagogy and technology. Faculty employed humor, ice breakers, and low stakes assessments to increase engagement and build group cohesion, thereby establishing a safe learning environment. Students' self-recorded skills practice using family members to play patient roles helped break down barriers and engage family, roommates, and pets in ways that increased faculty understanding of student contexts and home life. In turn, faculty teaching from their homes exhibited increased vulnerability and diminished power differentials as their homes and families were also on display. Despite physical distance, the pandemic, in some ways, brought students and faculty closer together.

Teaching presence refers to the organizational structure, the design, facilitation, and direction of cognitive and social processes to realize personally meaningful and education-

ally worthwhile learning outcomes [45]. Teaching presence depends on the instructor's ability to communicate goals and learning activities, motivate and engage the student and provide timely feedback [45]. Like previous research on online teaching presence [46], this study revealed a rapidly evolving organizational structure, learning activities, and teaching and assessment strategies. Faculty leveraged new virtual platforms and devised new active methods of teaching and assessment, including low stakes assessments on quiz applications, case scenarios, and group discussions.

As faculty were embracing new instructional platforms, they were also seeking resources at their workplaces. Unlike faculty dealing with limited access to technology [3] and reliable internet services [40], faculty in this study did not report challenges with infrastructure, although they had concerns about student internet connectivity. Without the technological tools that made remote instruction possible, higher education would have been significantly disrupted in the face of the pandemic [44]. However, rapid adoption of new technology came at a cost of stress, time, and feelings of inadequacy. Faculty were accustomed to being perceived as high functioning classroom educators. Indicators of course quality such as student satisfaction are heavily influenced by the extent to which educators are prepared to teach in the online learning environment [47]. Faculty across the three countries embraced resources and learning opportunities from the teaching and learning centers, librarians, and information technology services in their institutions. Faculty created high expectations for themselves to deliver an educational product equivalent to their classroom instruction, and some questioned if they expected too much of themselves in an unprecedented time. Despite the exhaustion, anxiety, and challenges, upon reflection, faculty were proud of their grit and resilience. Faculty reported overcoming many barriers to educating via a virtual platform and cited collaboration with fellow faculty as their primary resource for success.

Cognitive presence is the extent to which learners can construct and confirm meaning through sustained reflection and discourse in a community of inquiry [45]. Cognitive presence relates to the ability to obtain meaning through discussion and interaction with the community. Faculty built a cognitive teaching presence by developing new learning activities that could help students develop their knowledge and skills in a virtual medium [45]. Development of videos to demonstrate psychomotor and communication skills replaced in-person demonstrations. Increased virtual office hours were implemented to allow further co-construction of knowledge and replace organic conversations less likely to occur in the virtual medium. More frequent ungraded homework and pausing and questioning throughout the class were thoughtfully added to build depth and breadth of content delivered asynchronously. We noted that despite not having formal training in the community of inquiry framework, faculty from all three countries leveraged social, cognitive, and teaching presence.

The cognitive apprenticeship model (CAM) focuses on completing tasks in an authentic environment with modeling of expert reasoning and meaningful feedback as tasks are completed. The CAM focuses on using modeling, scaffolding, and coaching to help students integrate cognitive and metacognitive practices, along with articulation and reflection to promote problem-solving [27]. Modeling includes student learning through observation of experts. In the online environment, this may be achieved through videos that mimic real-life situations and role-modeling based on observing peers. Scaffolding entails supporting students in task execution through instructional, sequential modules, course rubrics, online discussions, and private communication. Coaching serves as a method to monitor student progress and activities, with support provided as needed. The instructor leverages videos, screencasts, and emails to interact and provide feedback [27]. Faculty in this study described using virtual patient experiences and virtual simulation to help students develop critical thinking skills and make connections to practice. Faculty used their home environments to create a meaningful context and facilitate conversation rather than lecture. Educators in some countries reported needing approval for telehealth from PT governing bodies. However, once approved, faculty leveraged telehealth and virtual

patient interactions and panels. Students interacting with patients in their homes and understanding patient context provided meaning and application of classroom learning when clinical experiences were limited due to the pandemic.

The pandemic provided a unique opportunity to practice alternate models of delivering physical therapist education. The physical therapist educators in this study acknowledged new ways of teaching online that they plan to bring forward to their onsite teaching, such as flipped classrooms, low stakes assessments, video practice of psychomotor skills, virtual simulation, and telehealth to access more diverse patient populations. However, although participants felt there are foundational and didactic aspects of PT curricula that may be suited to the online environment, ultimately, they described that the hands-on and communication skills unique to PT curricula are best practiced in an in-person environment.

*Limitations*

The authors recognize the limitations of the study. Sampling from institutions in three countries potentially limits the transferability of the results to the countries studied; however, we attempted to increase trustworthiness by using multiple cases in each context. Although 3–8 faculty members from each academic program also limits transferability, we found reemerging themes in the interviews and approached data saturation. While there was a greater representation of U.S faculty than faculty from Cyprus and Brazil, re-emerging themes were identified in data from those institutions also despite smaller samples. Two researchers from the study team completed independent coding and then reached intercoder agreement on codes and themes. While both researchers were from one institution in the U.S., researchers from Brazil and Cyprus agreed that the codes and themes developed represented the data gathered from their institutions. Future research may benefit from external auditor(s) given the researchers' roles in the academic programs. An additional limitation is that interviews from Cyprus and Brazil were translated into English by the researchers. Although a formal forward and backward translation process was not utilized for the translation of transcripts to English, both translators from Cyprus and Brazil were fluent in both languages, and care was taken to ensure that meaning was preserved in the translated transcripts.

## 5. Conclusions

Physical therapist faculty from three international PT programs describe teaching during the COVID-19 pandemic as one of the most challenging experiences of their professional careers. Despite this, faculty innovated and developed meaningful learning experiences for their students creating social, cognitive, and teaching presence. The virtual medium created challenges to student engagement and assessment of student understanding of content, and faculty found themselves working longer and harder to make authentic connections with students to ensure that understanding was achieved. Collaboration with faculty peers sharing this experience was perceived as the greatest facilitator of success. Ultimately, faculty persevered during a trying time due to the joy of contributing to the development of new health professionals. As institutions of higher education reopen for in-person learning [48], faculty have learned alternative approaches to delivering content that will enhance teaching practice in the future regardless of the context. Future research will explore the physical therapist student experience with online learning.

**Author Contributions:** Conceptualization, B.B.K. and K.N.; Formal analysis, L.P., B.B.K., C.P.K., and K.N.; Investigation, L.P., B.B.K., C.P.K., and K.N.; Methodology, L.P., B.B.K., C.P.K., K.N., S.G., and N.C.; Writing—original draft, L.P., B.B.K., C.P.K., K.N., S.G., R.M., and N.C.; Writing—review & editing, L.P., B.B.K., C.P.K., K.N., S.G., R.M., and N.C. All authors have read and agreed to the published version of the manuscript.

**Funding:** This research received no external funding.

**Institutional Review Board Statement:** Ethical approval granted from Mass General Brigham Institutional Review Board (Protocol #: 2020P002552), University Center Christus—Unichristus IRB (Protocol # 4.393.204), European University of Lefke University Ethics Committee (protocol # ÜEK/54/02/091920/02).

**Informed Consent Statement:** Informed consent was obtained from all subjects involved in the study.

**Conflicts of Interest:** The authors declare no conflict of interest.

**Appendix A**

1. Tell me what it is like to be a physical therapy educator in (Cyprus, United States) using remote instruction to teach physical therapist students online during the COVID-19 pandemic?
2. What resources were available to you to facilitate the transition to remote instruction?
3. What stories can you share about the experience that were particularly memorable?
4. Tell me some stories about how you navigated this challenge. Follow up questions:

    a. You said it was a challenge to . . . Can you say more about this challenge?

    b. You said that you had difficulty with . . . Can you elaborate?
5. What type of assessment methods did you use to evaluate students during the period of remote instruction?
6. Are there lessons learned from remote instruction that will inform your teaching practice moving forward when you return to the classroom?
7. Do you have anything else to share about teaching online during the COVID-19 pandemic?

    We are almost done with this interview. I am going to ask a few additional questions about you and your institution.

8. How many years have you been an educator?
9. Did you have experience with online learning or instruction prior to the COVID19 pandemic?

    a. Follow up question: Can you tell me more about your experience with online learning/instruction?
10. On average, how many physical therapist students are in your class?
11. What year of study are the students in (that you teach)?
12. Describe a typical semester including what a typical day looked like prior to the COVID-19 pandemic
13. What faculty supports are typically available on campus to help with teaching (prior to COVID19)?
14. In what ways did your Institution support you during the transition to remote instruction?

    Probing questions during the interview

- Could you say something more about . . . ?
- Can you give a more detailed description of . . . ?
- Can you think of times when . . . ?
- Do you remember a time when you noticed . . . .

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
