# Peer review of "Teaching Online during the COVID-19 Pandemic: A Phenomenological Study of Physical Therapist Faculty in Brazil, Cyprus, and The United States"

_education, doi:10.3390/educsci11030130_

Round 1

Reviewer 1 Report

The paper is very interesting and, of course, very topical. Congratulations. However, I believe there are some considerations that could help improve it.

- Firstly, it would be necessary to justify why these three universities have been selected: are the curricula similar? Could the results be conditioned by the selection of this sample?

- Secondly, in the procedure and analysis sections, triangulation has been mentioned, and that is correct, but, I believe what is missing is the inter- and intra-personal concordance analysis when establishing those codes and the subsequent content analysis. Has the kappa index been obtained? How many interviews were used for this?

- Thirdly, the results are supported by literal extracts from the conducted interviews. I believe that these are examples from the different universities involved. In this sense, it would be interesting to incorporate a coding system that would make it possible to understand where these reflections have been reported from in order to assess that the three fields have actually been taken into account when drawing conclusions. This is more a question of form than of content, as I believe that the procedure is correct.

Finally, the study raises two doubts based on the degree from which it has been approached. On the one hand, how has it been possible to redirect practical training in an area where it is so necessary? On the other hand, have you considered interviewing students to find out their opinion?

Author Response

Thank you for your time and your feedback. Please see attachment. 

Reviewer 2 Report

The study is original and addresses a current problem such as the effort of many teachers to adapt face-to-face teaching to virtual methods in times of COVID 19.
The objective of the study aims to know the barriers and facilitators to move from face-to-face teaching to virtual models, taking the perceptions of PT trainers in three different countries and contexts (USA, Brazil, Cyprus). Thus, it is understood that the objective tries to find out similarities and potential differences between countries. This differentiating aspect between countries is slightly reflected in the data collection and presentation, but not in the data analysis system.
Regarding the coding process established in five phases, it is necessary to specify how each researcher carried out the initial coding (steps 1 and 2); It is also advisable to clarify the coding strategy adopted as well as the types of codes that they applied (the reading of Saldaña on the coding process in qualitative research is recommended). It is necessary to substantiate the established initial coding process by means of bibliographic references.
It is also essential to describe in some detail how the joint search process for topics or categories was carried out, as well as the agreement between coders (steps 3,4,5). In addition, it is recommended to present the result of this joint decision in the form of a table or table detailing the transition from the initial codes to the final themes or categories agreed by the researchers along with the definitions of these categories found.
Considering the objective of the study from the three country contexts, it would have been interesting to have applied a coding strategy that would allow differentiating between the three countries and training realities, thus looking for differences or similarities in a more analytical and accurate way. Some authors such as Saldaña foresee this peculiarity by establishing a coding protocol and various types of codes to achieve similar ends to that proposed.

A thorough revision and new writing, more complete and detailed, of the analysis strategy of the qualitative data collected is recommended, thus being able to verify and justify the results presented, the discussion carried out and the proposed conclusions.

Author Response

Thank you for your time and feedback. Please see attached. 

Reviewer 3 Report

Thank you for the opportunity to read an interesting text on e-learning in times of pandemic. The authors did a comparative study with one tool in three countries: Brazil, USA and Northern Cyprus. This is a very unusual comparison due to the different geographical locations of the countries. Moreover, the study seems interesting for another reason as well. The authors showed the characteristics of the training process of physiotherapy specialists. This is a unique area requiring the formation of practical skills and in-depth knowledge. I have only a few comments on the text.

  1. In the introduction, it is worth making an even more precise argument the necessity of comparative studies in three countries. The reader must see the logical justification for the inclusion of the USA, Cyprus and Brazil.
  2. As part of the formulation of the theoretical framework, it is worth adding information on the differences resulting from crisis e-learning versus e-learning implemented with full methodical principles. It is worth showing in the introduction in which mode the classes are implemented. Whether academic teachers, specialists - practitioners conducting classes were prepared for this form in all three countries.
  3. The sampling procedure is not entirely clear. Has there been saturation of the research sample? Are the variables that emerged during the analyses also saturated?
  4. In the discussion or the empirical part, it is useful to show, e.g. in the form of a diagram or a table, the similarities and differences in the individual countries. The results must have a clear summary also for the comparative part of this study.
  5. Do not include quotations in the discussion section (p.10). Such elements should be in the empirical section. 

I keep my fingers crossed for making additions and minor corrections.

Author Response

(The authors gave the same response as above.)

Round 2

Reviewer 2 Report

A better treatment of the data analysis is observed through the presentation of the coding procedure.

The coding system is adequate, respecting the scientific procedure by structuring the study in two coding processes, initial and final.

The table resulting from the final codes together with the descriptions is positively valued.

The presentation of results is better understood and the discussion makes more sense.

The study in general is coherent, logical, meticulous and technically better developed.

The treatment of the bibliography with the changes made is also adequate.